# An Enantioselective Approach to 4-Substituted Proline Scaffolds: Synthesis of (*S*)-5-(*tert*-Butoxy carbonyl)-5-azaspiro[2.4]heptane-6-carboxylic Acid

**DOI:** 10.3390/molecules25235644

**Published:** 2020-11-30

**Authors:** Blanca López, Martí Bartra, Ramon Berenguer, Xavier Ariza, Jordi Garcia, Roberto Gómez, Hèctor Torralvo

**Affiliations:** 1Departament de Química Inorgànica i Orgànica, Secció de Química Orgànica, Facultat de Química, Universitat de Barcelona, Martí i Franquès 1-11, 08028 Barcelona, Spain; blunch0@hotmail.com (B.L.); rogogar@gmail.com (R.G.); hectortorralvo@ub.edu (H.T.); 2R&D Department, Esteve Química S.A., Caracas 17-19, 08030 Barcelona, Spain; mbartra@eqesteve.com (M.B.); rberenguer@eqesteve.com (R.B.); 3Institut de Biomedicina (IBUB), Universitat de Barcelona, 08028 Barcelona, Spain; 4CIBER Fisiopatología de la Obesidad y la Nutrición (CIBERobn), Instituto de Salud Carlos III, 28029 Madrid, Spain

**Keywords:** phase-transfer catalysis, stereoselective synthesis, antiviral agent, proline analogue

## Abstract

A catalytic and enantioselective preparation of the (*S*)-4-methyleneproline scaffold is described. The key reaction is a one-pot double allylic alkylation of an imine analogue of glycine in the presence of a chinchonidine-derived catalyst under phase transfer conditions. These 4-methylene substituted proline derivatives are versatile starting materials often used in medicinal chemistry. In particular, we have transformed *tert*-butyl (*S*)-4-methyleneprolinate (**12**) into the *N*-Boc-protected 5-azaspiro[2.4]heptane-6-carboxylic acid (**1**), a key element in the industrial synthesis of antiviral ledipasvir.

## 1. Introduction

Non-natural amino acids are often used as building blocks of active pharmaceutical compounds [1,2,3,4]. In particular, proline analogues are especially attractive since their presence in peptide chains causes conformational restrictions which are critical for biological activity [3]. Among modified prolines, the Boc-protected amino acid 5-azaspiro[2.4]heptane-6-carboxylic acid (**1**, Figure 1) has recently become a key element in the synthesis of ledipasvir (**2**) [5], a potent non-structural protein 5A (NS5A) inhibitor that is used for treatment of hepatitis C virus infections (HCV) [6,7,8].

More than 110 million people are infected with HCV worldwide, of which 700,000 die annually and chronic infection affects another 80 million. Progression of the disease often leads to cirrhosis or hepatic carcinoma.

Ledipasvir (**2**) is one of the more effective known anti-viral agents with an activity in the range of picomolar concentrations and, in combination with other antivirals, it is widely used for this infection.

The spirocyclic structure of Ledipasvir is always synthesized enantiomerically pure, either by resolution of the racemic mixture [5,9,10,11,12] or from enantiopure natural amino acid derivatives: pyroglutamic acid (**4**) [5,13] or 4-hydroxyproline (**3**) [5,14,15,16] (see Scheme 1) [17,18]. Nevertheless, an enantioselective and catalytic approach to 4-substituted proline scaffold is still lacking. Herein, we describe an easy and high-yielding process for the preparation *N*-Boc-(*S*)-4-methyleneproline (**5**) (or its *t*-Bu ester, **6**), useful precursors of different 4-substituted prolines. Indeed, these compounds (*S* forms) have previously been used in medicinal chemistry [19,20] and in the industrial development of new HCV protease inhibitors [21]. On the other hand, compound **5** is a suitable precursor of *N*-protected amino acid **1** [5].

The preparation of enantioenriched α-amino acid derivatives by alkylation of benzophenone imines of glycine alkyl esters under chiral phase-transfer catalysis (PTC) was first introduced by O′Donnell in 1978 [22]. Subsequent years witnessed impressive improvements reported by the groups of Lygo [23,24] and Maruoka [25], among many others. Nowadays, chiral phase-transfer catalysis is a well-established strategy that has been applied to a large number of synthetic goals using chiral natural or non-natural derived catalysts [22,26,27,28].

## 2. Results and Discussion

We envisaged that the direct enantioselective alkylation of glycine-derived imines with 1,1′-bis(halomethyl)cyclopropanes under PTC conditions would give ready access to the 4-spiro cyclopropane proline backbone. Unfortunately, the neopentylic nature of the alkylating reagent prevented any alkylation reaction. Thus, 1,1′-bis(bromomethyl)cyclopropane, 1,1′-bis(iodomethyl)cyclopropane (and also 1,1′-bis(mesityloxy methyl)cyclopropane) did not react with neither *tert*-butyl *N*-diphenylmethyleneglycinate (**7**) nor *p*-chlorophenylmethylene imine **8** in the presence of a commercially available chinchonidine-derived [26,27] catalyst **9** (Scheme 2) using different bases (KOH or CsOH) and solvents (toluene or CH_2_Cl_2_) under phase-transfer conditions [26].

Looking for improved reactivity, we turned our attention to the more activated bis allylic alkylating systems (Scheme 3). The reaction of imine **7** with an excess of 3-bromo-2-(bromomethyl)-1-propene (**10**) in the presence of catalyst **9** (10 mol%) afforded an interesting mono-alkylated product **11**. Unexpectedly, we observed that **11** was converted in situ in the NMR tube to the desired methylene proline **12** in 40% overall yield on standing overnight at room temperature.

Encouraged by this promising result, the same sequence was performed, but without isolating alkylated intermediate imine **11**. Furthermore, CDCl_3_ was substituted by non-deuterated chloroform in the cyclization. As a result, the product was obtained in a better yield but the excess of alkylating agent **10** promoted a further alkylation of the desired product **12** and dimeric proline **13** (Figure 2) could be obtained (up to 10% yield) when 5 equivalents of **10** was used.

Several conditions for the initial alkylation were explored. Best yields were obtained when a slight excess of dibromide **10** was used in mixtures of toluene/CH_2_Cl_2_ and KOH as a base. Reaction temperature affected yield and enantioselectivity. Yield decreases dramatically below –20 °C, whereas a significant enantioselectivity loss was observed above 0 °C. The highest yield of **12** (71%) was obtained with 2.5 equivalent of **10** at –20 °C for 7 h (Scheme 4). Under these conditions, enantioselectivity was determined by HPLC analysis of the *N*-benzyloxycarbonyl derivative of **12** and showed a 95:5 enantiomeric ratio (e.r.). Under the same conditions, slightly worse enantioselectivity (93:7 e.r.) and lower yield (38%) was obtained when 4 equivalents of **10** were used. Regarding intramolecular alkylation, it can be carried out in dichloromethane but chloroform was a more effective solvent [28].

With enantioenriched **12** in hand, the nitrogen atom was protected with the Boc group to afford protected amino acid **6**. Basic hydrolysis of **6** easily afforded *N*-Boc 4-methylene proline **5** without loss of enantiopurity (checked by HPLC analysis of the corresponding benzyl ester). As far as we know, this is the first catalytic enantioselective preparation of these unsaturated prolines [29].

Subsequently, we proposed the preparation of amino acid **1**. Direct cyclopropanation using the classical Simmons–Smith reaction (ZnEt_2_, CH_2_I_2_) of **12**, **6** or **5** led to complex mixtures with significant amounts of deprotected products, probably due to the necessary acidic media [14]. Better results were achieved by addition of dibromocarbene species to the double bond followed by debromination. Several sequences of cyclopropanation/debromination/deprotection procedures were explored based on the results reported for Me esters [15] (or the free acid) [9] analogues in the literature (Scheme 5).

As far as cyclopropanation is concerned, better yields (73%) were obtained in the cyclopropanation of ester **6** with CBr_3_CO_2_Na rather than in the addition of dibromocarbene arising from bromoform on the corresponding carboxylic acid **5** (40%). Dibromo derivatives (from **5** and **6**) were not isolated but were used as a mixture of diastereoisomers in the next step. In the debromination step, hydrogenolysis gave better results than radical reduction with tris(trimethyl)silane (83% vs. 53%). Therefore, according to our results, the best way to synthesize target molecule **1** is: (i) dibromopropanation of **6** to afford a crude mixture of diastereoisomers, (ii) ester deprotection under basic conditions to afford dibromo acids (80% yield) and, (iii) final debromination with hydrogen to furnish only one stereoisomer. The overall yield of the sequence was 49%. The enantiopurity of Boc-amino acid **1** was checked by chiral HPLC of the corresponding benzyl ester.

## 3. Materials and Methods

### 3.1. General Considerations

Compounds **7**–**10** are commercially available. ^1^H (400 MHz) and ^13^C (101 MHz) nuclear magnetic resonance (NMR) spectra were registered using a Varian Mercury 400 MHz NMR spectrometer. CDCl_3_ (99.9%) was used as solvent while SiMe_4_ (TMS) was used as reference. Coupling constants (*J*) are expressed in Hz and chemical shifts in parts per million (ppm). Signal multiplicities are reported using the following abbreviations: s (singlet), d (doublet), t (triplet), q (quadruplet), m (multiplet), and br (broad). More complicated signals are described as a combination of the indicated abbreviations: e.g., dd (doublet of doublets), dt (doublet of triplets). Two-dimensional experiments (NOESY, HSQC) were performed to confirm the signal assignments. ESI (+) or (–) spectra were acquired either on an LC/MSD-TOF instrument or on a ZQ mass spectrometer, utilizing a mixture of H_2_O:CH_3_CN (1:1, *v*/*v*) as the eluent.

### 3.2. Synthesis of Compound ***12***

Dibromide **10** (1.82 g, 8.5 mmol) was dissolved in a mixture of toluene (3.5 mL) and CH_2_Cl_2_ (1.5 mL) and was added to a solution of imine **7** (1.00 g, 3.4 mmol) and catalyst **9** (176 mg, 0.34 mmol) in toluene (7 mL) and CH_2_Cl_2_ (3 mL) at –20 °C. Aqueous KOH (50%, 7.6 mL) was added and the mixture was stirred for 7 h at –20 °C. The phases were separated, and the organic layer was dried over anhydrous MgSO_4_. After filtration, the solvent was evaporated, CHCl_3_ (10 mL) was added and the solution was stirred overnight at rt. The solution was extracted with water (3 × 15 mL). The combined aqueous layer was basified with aqueous NaOH until pH = 11 and extracted with CH_2_Cl_2_ (2 × 10 mL) and AcOEt (2 × 10 mL). The combined organic layer was dried over anhydrous MgSO_4_. After filtration, the solvent was evaporated to yield an oil. Flash chromatography (SiO_2_, hexanes/AcOEt 9:1) of the crude afforded **12** as an oil (445 mg, 71%). By-product **13** (36 mg, 5%) was also isolated. An analytical sample of **12** was transformed into the *N*-benzyloxycarbonyl derivative. This derivative was analyzed by HPLC with a CHIRALPAK^®^ IA (0.46 cm × 25 cm) column using a 90:10 hexanes/isopropanol mixture as eluent (6.24 min, *R* isomer; 6.79 min, *S* isomer) giving a 95:5 enantiomeric ratio (e.r.).

**12**: colorless oil. [α]_D_^25^ –20.4 (*c* 1.0, MeOH). ^1^H NMR (400 MHz, CDCl_3_): δ 4.95 (m, 1H), 4.92 (m, 1H), 3.75 (m, 1H), 3.69 (d, *J* = 16.0 Hz, 1H), 3.50 (d, *J* = 16.0 Hz, 1H), 2.81–2.75 (m, 1H), 2.56–2.46 (dd, *J* = 16.0, 4.0 Hz, 1H), 2.33 (br s, 1H), 1.46 (s, 9H). ^13^C{^1^H} NMR (101 MHz, CDCl_3_): δ 173.6, 147.9, 105.4, 81.4, 61.1, 51.3, 37.0, 28.1. IR (ATR, cm^–1^): 2974, 2929, 1725, 1658, 1366, 1145, 702. HRMS (ESI): (*m*/*z*) [M + H]^+^ calcd. for C_10_H_18_NO_2_ 184.1338, found 184.1334.

**13**: yellowish oil. ^1^H NMR (400 MHz, CDCl_3_): δ 5.09 (s, 2H), 4.88 (m, 4H), 3.65 (m, 2H), 3.32 (m, 4H), 3.14 (m, 4H), 2.68 (m, 2H), 2.55 (m, 2H), 1.45 (s, 18H). ^13^C{^1^H} NMR (101 MHz, CDCl_3_): δ 172.8, 146.5, 144.0, 114.4, 105.3, 80.9, 66.2, 57.8, 57.3, 36.7, 28.3. IR (ATR, cm^−1^): 2974, 2927, 1735, 1723, 1366, 1275, 1144, 701. HRMS (ESI): (*m*/*z*) [M + H]^+^ calcd. for C_24_H_39_N_2_O_4_ 419.2910, found 419.2905.

### 3.3. Synthesis of Compound ***6***

Boc_2_O (537 mg, 2.46 mmol) was added to a solution of amine **12** (410 mg, 2.23 mmol), anhydrous Et_3_N (0.62 mL, 4.46 mmol), DMAP (27 mg, 0.223 mmol) in anhydrous CH_2_Cl_2_ (5 mL) at rt under N_2_. The mixture was stirred for 48 h. Aqueous HCl (1 M, 5 mL) was added. The organic layer was extracted with brine and dried over anhydrous MgSO_4_. The solvent was evaporated to obtain compound **6** as an oil (575 mg, 85%).

**6** [30]: oil. [α]_D_^25^ –21.92 (*c* 1.00, CHCl_3_) [lit. [30] [α]_D_^25^ –20.9 (*c* 1.01, CDCl_3_)]. ^1^H NMR (400 MHz, CDCl_3_, mixture of rotamers, minor rotamer*): δ 5.00 (m, 2H), 4.36* (m, 1H), 4.27 (m, 1H), 4.05 (m, 2H), 3.02–2.83 (m, 1H), 2.56 (d, *J* = 16.0 Hz, 1H), 1.48* (s, 9H), 1.46 (s, 9H), 1.45 (s, 9H). ^13^C{^1^H} NMR (101 MHz, CDCl_3_): δ 172.3, 171.7*, 154.9*, 154.5, 144.5*, 143.8, 108.2, 107.9*, 81.8, 80.5, 60.9*, 60.3, 51.4, 37.5, 36.7*, 29.0, 28.9*, 28.5, 21.6, 14.8. IR (ATR, cm^−1^): 2923, 2853, 1725, 1710, 1367, 1049. HRMS (ESI): (*m*/*z*) [M + Na]^+^ calcd. for C_15_H_25_NNaO_4_ 306.1681, found 306.1681.

### 3.4. Synthesis of Compound ***5***

Ester **6** (436 mg, 1.54 mmol) was stirred in a mixture of MeOH (10 mL), THF (5 mL), H_2_O (5 mL) and aqueous KOH (5 mL, 50%) at rt for 32 h. The organic solvents were evaporated and the residue was acidified with HCl 2 M to pH = 2. AcOEt (20 mL) was added, the organic layer was separated, and the aqueous layer was extracted with AcOEt (2 × 20 mL). The combined organic layer was dried over anhydrous MgSO_4_ and the solvent was evaporated to afford acid **5** as an oil (349 mg, 78%).

**5** [31]: oil. [α]_D_^25^ –27.7 (*c* 1.00, CHCl_3_). ^1^H NMR (400 MHz, CDCl_3_, mixture of rotamers, minor rotamer*): δ 10.73 (br s, 1H), 4.98 (br s, 2H), 4.36–4.50 (m, 1H), 4.06 (m, 1H), 4.00* (bs, 1H), 2.89–3.0 (m, 1H), 2.65–2.74 (m, 1H), 1.44* (s, 9H), 1.39 (s, 9H). ^13^C{^1^H} NMR (101 MHz, CDCl_3_, minor rotamer*): δ 177.4*, 176.7, 154.9*, 154.0, 142.9*, 142.0, 108.1, 108.0*, 80.7, 77.3*, 58.9, 58.5*, 50.8*, 50.4, 36.6, 35.6*, 28.3*, 28.1. HRMS (ESI): (*m*/*z*) [M-H]^–^ calcd. for C_11_H_16_NO_4_ 226.1079, found 226.1084.

A catalytic sample of **5** was transformed into their benzyl ester derivative: A mixture of **5** (10 mg, 0.04 mmol) and Cs_2_CO_3_ (14 mg, 0.04 mmol) was stirred under nitrogen in anhydrous DMF (0.5 mL) at rt. Benzyl bromide (5 µL, 0.04 mmol) was added and the mixture was stirred overnight at rt. Aq. sat. NaHCO_3_ sol. (2 mL) was added. After 5 min, the aqueous phase was extracted with DCM (3 × 2 mL), dried over MgSO_4_ and filtered. The solvent was evaporated and the crude was purified by column chromatography (SiO_2_, hexanes:EtOAc 90:10) to afford the pure benzyl ester as a yellowish oil (11 mg, 79%). HPLC analysis with a CHIRALPAK^®^ IC (0.46 cm × 25 cm) column using a 92:8 hexanes/isopropanol mixture as eluent (1 mL/min) gave 95:5 enantiomeric ratio (e.r.). (9.27 min, major isomer; 9.97 min, minor isomer).

An analytical sample of benzyl ester or **5** for HPLC analysis purposes was prepared: LiHDMS (1 M in THF, 0.07 mmol) was added to a sample of enantioenriched benzyl ester from **5** (10 mg, 0.035 mmol) in anhydrous THF (1 mL) at –78 °C under N_2_. After 10 min, the solution was warmed to 0 °C and quenched with pH 7 buffer solution (1 mL). The mixture was partitioned with DCM (2 mL) and the aqueous layer was extracted with additional DCM (2 × 2 mL). The combined organic phases were dried over anhydrous MgSO_4_. The solvent was evaporated to obtain a crude benzyl ester. Flash chromatography (SiO_2_, hexanes:EtOAc 90:10) was used to yield the pure racemic benzyl ester as a colorless oil (8 mg).

### 3.5. Synthesis of Compound ***1***

CBr_3_CO_2_Na (2.230 g, 7.00 mmol) was added to a solution of compound **6** (900 mg, 3.18 mmol) and Bu_4_NBr (21 mg, 0.064 mmol) in anhydrous CH_2_Cl_2_ (15 mL) at rt under N_2_. The mixture was heated in an oil bath at 70 °C for 2.5 h and an additional amount of CBr_3_CO_2_Na (304 mg, 0.954 mmol) was added and the reaction was heated at 70 °C for an additional 3 h. A solution 7:3 of *tert*-butyl methyl ether and hexane (20 mL) was added at rt. The mixture was filtered through Celite^®^. The organic layer was washed with water (20 mL) and brine (20 mL), and then dried over anhydrous MgSO_4_. The solvent was evaporated and the residue was filtered through a short silica column (90:10 hexanes/AcOEt) to obtain crude dibromo *tert*-butyl ester as an yellowish oil (6:4 mixture of diastereomers, 1.056 g, 73%). HRMS (ESI) (*m*/*z*) [M + Na]^+^calcd. for C_16_H_25_Br_2_NNaO_4_ 476.0048, found 476.0047.

A sample of the crude *tert*-butyl ester (892 mg, 1.96 mmol) was stirred in a mixture of MeOH (20 mL), THF (10 mL), H_2_O (10 mL) and aqueous KOH (10 mL, 50%) at rt for 48 h. The organic solvents were evaporated and the residue was acidified with HCl 2M to pH = 2. AcOEt (30 mL) was added, the organic layer was separated, and the aqueous layer was extracted with AcOEt (2 × 25 mL). The combined organic phases were dried over anhydrous MgSO_4_, the solvent was evaporated and the mixture was filtered through SiO_2_ (CH_2_Cl_2_:MeOH 90:10) to afford the dibromo acid as a brownish solid (80%). HRMS (ESI): (ESI): (*m*/*z*) calcd. for [M-H]^–^: 395.9452, found: 395.9456.

The dibromo acid (620 mg) and KOH (554 mg, 30 mmol) in isopropanol (15 mL) were stirred under N_2_ until the mixture became homogeneous. An amount of 10% Pd/C (132 mg, 0.122 mmol) was added and the resulting mixture was stirred in an oil bath at 40 °C under an H_2_ atmosphere for 48 h. The cold mixture was filtered through Celite^®^ and the pad was washed with water (20 mL). The organic solvents were removed and *tert*-butyl methyl ether (20 mL) and 2M aq. HCl (20 mL) were added to the remaining solution. The phases were decanted and the aqueous phase was further washed with *tert*-butyl methyl ether (2 × 20 mL). The combined organic phases were dried over anhydrous MgSO_4_. The solvent was evaporated to obtain crude compound **1**. Flash chromatography (SiO_2_, 95:5 CH_2_Cl_2_/MeOH) of the crude afforded **1** as a solid (311 mg, 83%).

(*S*)-5-(*tert*-butoxycarbonyl)-5-azaspiro[2.4]heptane-6-carboxylic acid (**1**).[14] Melting point: 94–5 °C. [α]_D_^25^ –23.5 (*c* 1.00, MeOH) [31]. ^1^H NMR (400 MHz, CDCl_3_, rt, 6:4 mixture of rotamers, minor rotamer*): δ 10.64 (br s, 1H), 4.50 (m, 1H), 4.39* (m, 1H), 3.40–3.39 (m, 3H), 3.20* (m, 1H), 2.29 (m, 1H), 1.95 (m, 1H), 1.48* (s, 9H), 1.44 (s, 9H), 0.61 (m, 4H). ^13^C{^1^H} NMR (101 MHz, CDCl_3_, rt, mixture of rotamers, minor rotamer*): δ 178.6, 176.1*, 155.8*, 154.0, 81.1*, 80.5, 59.5, 54.4*, 53.7, 39.0, 37.2*, 28.5, 28.4*, 20.7*, 20.2, 13.2*, 11.8, 9.4, 8.0*. IR (ATR): 2967, 2929, 2872, 1717, 1625, 1432, 1365, 1176, 1144, 1116. HRMS (ESI): (*m*/*z*) [M-H]^–^ calcd. for C_12_H_18_NO_4_ 240.1236, found 240.1236.

A sample of **1** [32] was transformed into their benzyl ester derivative: A mixture of **1** (100 mg, 0.41 mmol) and Cs_2_CO_3_ (135 mg, 0.41 mmol) was stirred under nitrogen in anhydrous DMF (1 mL) at rt. Benzyl bromide (49 µL, 0.41 mmol) was added and the mixture was stirred overnight at rt. Aq. sat. NaHCO_3_ sol. (5 mL) was added. After 10 min, the aqueous phase was extracted with DCM (3 × 5 mL), dried over MgSO_4_ and filtered. The solvent was evaporated to obtain 124 mg of the crude benzyl ester. The crude was purified by column chromatography (SiO_2_, hexanes:EtOAc 90:10) to afford the pure benzyl ester as a colorless oil (114 mg, 82%).

Benzyl ester of **1**. Colorless oil. [α]_D_^25^ –15.0 (*c* 1.00, MeOH). ^1^H NMR (400 MHz, CDCl_3_, rt, 6:4 mixture of rotamers, minor rotamer*): δ 7.35 (m, 5H), 5.27* (d, *J* = 12 Hz, 1 H), 5.16 (AB system, 2 H), 5.08* (d, *J* = 12 Hz, 1 H), 4.53* (dd, *J* = 8 Hz, 4 Hz, 1H), 4.42 (dd, *J* = 8 Hz, 4 Hz, 1H), 3.42–3.24 (m, 3H), 3.25* (m, 1H), 2.27 (m, 1H), 1.80 (dd, *J* = 12 Hz, 4 Hz, 1H), 1.73* (dd, *J* = 12 Hz, 4 Hz, 1H), 1.45* (s, 9H), 1.36 (s, 9H), 0.38–0.67 (m, 4H). ^13^C{^1^H} NMR (101 MHz, CDCl_3_, rt, mixture of rotamers, minor rotamer*): δ 172.7, 172.4*, 154.3*, 153.7, 135.9*, 135.6, 128.5, 128.4, 128.3, 128.2*, 128.1*, 79.9, 79.8*, 66.6, 59.7, 59.3*, 54.0*, 53.6, 39.0, 38.2*, 28.4*, 28.2, 20.6*, 20.0, 12.9*, 12.2, 8.9, 8.3*. IR (ATR): 2974, 2869, 1749, 1698, 1396, 1170, 1113. HRMS (ESI): (*m*/*z*) [M + H]^+^ calcd. for C_19_H_26_NO_4_ 332.1862, found 332.1859.

HPLC analysis with a CHIRALPAK^®^ IC (0.46 cm × 25 cm) column using a 92:8 hexanes/isopropanol mixture as eluent (1 mL/min) gave a 95:5 enantiomeric ratio (e.r.) (12.85 min, major isomer; 14.57 min, minor isomer) (See Appendix A).

A sample of commercial enantioenriched compound 1 (*S* isomer, Fluorochem) was also transformed in the same way into the benzyl ester and used as a standard in HPLC experiments. Moreover, a sample of racemic benzyl ester was also prepared: LiHDMS (1 M in THF, 0.30 mmol) was added to a solution of benzyl ester arising from commercial **1** (94 mg, 0.30 mmol) in anhydrous THF (5 mL) at –78 °C under N_2_. After 10 min, the solution was warmed to 0 °C and quenched with pH 7 buffer solution (5 mL). The mixture was partitioned with DCM (10 mL) and the aqueous layer was extracted with additional DCM (2 × 5 mL). The combined organic phases were dried over anhydrous MgSO_4_. The solvent was evaporated to obtain a crude benzyl ester. Flash chromatography (SiO_2_, hexanes:EtOAc 90:10) was used to afford the pure racemic benzyl ester as a colorless oil (90 mg, 95%).

### 3.6. Synthesis of Compound ***14***

(Me_3_Si)_3_SiH (166 mg, 0.67 mmol) and AIBN (3.6 mg, 0.022 mmol) were added to a solution of the crude dibromo *tert*-butyl ester arising from **6** (101 mg, 0.22 mmol) in anhydrous toluene at 0 °C under an N_2_ atmosphere. The mixture was stirred and heated in an oil bath at 90 °C for 18 h. The solvent was removed and the crude was purified by flash chromatography (SiO_2_, 9:1 hexane/AcOEt) to afford **15** as an yellowish oil (34 mg, 53%).

**14** [9]: oil. [α]_D_^25^ –21.4 (*c* 1.00, CHCl_3_). ^1^H NMR (400 MHz, CDCl_3_, rt, mixture of rotamers, minor rotamer*): δ 4.34* (m, 1H), 4.25 (m, 0.7H), 3.44 (m, 1H), 3.37* (m, 1H), 3.30 (m, 1H), 3.23* (m, 1H), 2.36–2.25 (m, 1H), 1.71 (m, 1H), 1.50 (s, 1H), 1.46* (s, 1H), 1.45 (s, 1H), 0.63–0.45 (m, 4H). ^13^C{^1^H} NMR (101 MHz, CDCl_3_, rt, mixture of rotamers, minor rotamer*): δ 172.0, 171.9*, 154.3*, 154.0, 81.0, 79.7, 79.6, 60.5, 60.3*, 54.2*, 53.8, 39.4, 38.5*, 28.6, 28.5, 28.1, 20.7*, 20.0, 13.6, 8.4. IR (ATR, cm^−1^): 2975, 2929, 2866, 1743, 1698, 1391, 1365, 1146, 1110, 769. HRMS (ESI) (*m*/*z*) [M + H]^+^ calcd. for C_16_H_28_NO_4_ 298.2018, found 298.2020.

### 3.7. Hydrolysis of Compound ***14***

An aqueous solution of KOH (50%, 1 mL) was added to a solution of *tert*-butyl ester **14** (50 mg, 0.17 mmol) in MeOH (2 mL), THF (1 mL) and water (1 mL). The mixture was stirred 24 h at rt. The organic solvents were evaporated and the residue was acidified with HCl 2M to pH = 2. The mixture was extracted with AcOEt (10 mL), the organic layer was separated, and the aqueous layer was extracted with additional AcOEt (2 × 10 mL). The combined organic layer was dried over anhydrous MgSO_4_ and the solvent was evaporated to afford almost pure compound **1** (34 mg, 82%). M.p. 95–6 °C. [α]_D_^25^ –23.0 (*c* 1.00, MeOH).

### 3.8. Cyclopropanation of Compound ***5***

Aqueous NaOH (7.6 mL, 50%) was added dropwise to a solution of acid **5** (250 mg, 1.1 mmol), BnN(Me)_3_Cl (14 mg, 0.077 mmol) and CHBr_3_ (834 mg, 3.3 mmol) in toluene (3 mL) at rt. The mixture was heated in an oil bath to 60 °C for 28 h. Toluene was evaporated and CH_2_Cl_2_ (10 mL) was added. The organic layer was separated, washed with water and HCl 2M and dried over anhydrous MgSO_4_. After filtration through a short pad of silica, the solvent was evaporated to obtain a solid mixture of crude dibromide diastereomers (176 mg, 40%) which was used in the next step. Crude dibromide (176 mg) and KOH (148 mg, 0.80 mmol) were stirred in isopropanol at rt until the solid dissolved. An amount of 10% Pd/C (35 mg, 0.033 mmol) was added and the mixture was stirred under H_2_ atmosphere at 40 °C in an oil bath for 48 h. The suspension was cooled down and filtered through Celite^®^. The pad was washed with water (6 mL) and the volatiles were removed. The residue was partitioned by addition of 2M aq. HCl (6 mL) and *tert*-butyl methyl ether (6 mL). The aqueous phase was extracted with *tert*-butyl methyl ether (2 × 6vmL). The combined organic phases were dried over anhydrous MgSO_4_ and the solvent was evaporated to afford compound **1** (88 mg, 83%).

## 4. Conclusions

In summary, we described herein the first enantioselective and catalytic approach to the useful protected amino acids **1** and **5** as well as their methylene precursor **12**. The key step was the alkylation of the glycine-derived imine **7** with allylic dibromide **10** under phase transfer conditions in the presence of 10% of a commercially available chinchonidine-derived catalyst.

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
