# Peer review of "An Enantioselective Approach to 4-Substituted Proline Scaffolds: Synthesis of (S)-5-(tert-Butoxy carbonyl)-5-azaspiro[2.4]heptane-6-carboxylic Acid"

_molecules, 2020, doi:10.3390/molecules25235644_

Round 1
Reviewer 1 Report
The paper by López and co-workers is devoted to enantioselective synthesis of (S)-4-methyleneproline as an intermediate for the spiro derivative, a component of ledipasvir. The aim was achieved by alkylation of glycine-derived imine in the presence of cinchonidine catalyst. Cyclopropanation of the obtained compound was also described.
The described methodology is interesting, the paper is well written, experiments were properly designed and performed. My only concern is that part of the compounds (number 1, 5, 6, 14) are known and already described either in open literature or the patents. The appropriate citations are missing, and the spectral data and other physicochemical data are not compared with the previous preparations.
The file "non-published material" is identical as "supplementary file".
Author Response
As Reviewer 1 points out, compounds 1, 5, 6 and 14 have been reported previously in the literature. The appropriate citations for these compounds are now added in the experimental part in the revised manuscript.
Dr. Paul Lloyd-Williams, a native English speaking lecturer has reviewed the entire text.
Additions and corrections are marked in color
Reviewer 2 Report
The work contains interesting data, it is written in an interesting way (language, manner of presentation, scientific thread)). The synthesis and preparation recipes seem well optimized. The structure of the compounds was confirmed using modern spectroscopic methods. The compounds are characterized correctly. Generally, the chemical part (research presented) is correct.
Generally, the work is interesting study and should be published in Molecules in a present form.
Author Response
Thank you for your kind comments